# OpenReview forum: "Reference-Specific Unlearning Metrics Can Hide the Truth: A Reality Check"
_ICLR.cc/2026/Conference — Submitted to ICLR 2026_

### Official Review · Reviewer_1R1j · 2025-10-30

**Soundness:** 2
**Presentation:** 4
**Contribution:** 3
**Rating:** 4
**Confidence:** 4

**Summary:**

Overall, this is work provides keen insights and moves the machine unlearning field in the right direction. The observation that unlearning models should match gold standard retain models is well-motivated. The work is well-organized, and it fits well in the literature. My main concern is that it is unclear if choice of retain model affects the FADE score. The authors claim that FADE is more robust, but I did not find a sensitivity study which answers this question. Moreover, the authors observe that FADE is typically far better for the original model than for any unlearned model. While FADE appears useful for comparing unlearned models, it calls into question whether FADE is sensitive to behaviors of the retain model that aren’t important in practical scenarios. I believe this is a promising work, and I would be happy to adjust my score if my concerns are addressed.

The authors claim that the core objective of unlearning is that the unlearned model should behave indistinguishably from a retrained model and that current unlearning metrics do not accomplish this. The authors introduce FADE which measures the symmetric KL divergence of the retain model distribution and the unlearned model distribution given some condition. The proposed metric disagrees with current metrics on TOFU and UnlearnCanvas, indicating that current metrics are not good measures of equivalence with (gold standard) retrain models.
The authors find that forget quality (as measured by current metrics on TOFU) is sensitive to choice of reference answer, indicating that current metrics are brittle in evaluating unlearning performance.

**Strengths:**

Distributional measures of similarity between the unlearned model and a retrained model are a step in the right direction for unlearning metrics, and the authors’ insights regarding this are valuable

The paper is clearly written and easy to follow. It is well grounded in literature.

The observation that unlearning should move beyond static evaluations is valid and moves the field in the right direction.

The consistency of FADE in figure 5 is compelling (compared with differing behaviors of FQ)

**Weaknesses:**

It is not immediately clear why the distribution of a single unlearned model should fit the exact distribution of a single retrain model. Exact unlearning in the literature (Nguyen et al 2025 “A Survey of Machine Unlearning”) is the case where the distribution of retrain models matches the distribution of unlearned models. Wouldn’t it be more appropriate to measure the expected FADE (or something similar) over distributions of models?

There is no study of the robustness of FADE to choice of reference model. Is this more robust than FQ?

Questions
What is “functional alignment”? It seems to be matching output distributions or something to that effect.

Is FADE robust to choice of reference model (i.e., is it stable when you test it with different randomly initialized and retrained models)?

Line 122 “We expect to achieve more robust unlearning that better withstands such post-unlearning attacks.” If FADE measures how well unlearned models mimic a retained model, why would FADE help make unlearning methods more robust to post-unlearning attacks? My understanding is that the latent information remains inside the model and it is not revealed at the logits/output until after the attack.

What is the variance of FADE across multiple unlearned models compared to the same reference model?

One could argue that it shouldn’t matter if an unlearned model maps the target concept to reasonable concept A vs reasonable concept B, but this could significantly impact FADE. What is the variance of FADE in this case, and should these semantics matter in practical scenarios?

**Questions:**

What is “functional alignment”? It seems to be matching output distributions or something to that effect.

Is FADE robust to choice of reference model (i.e., is it stable when you test it with different randomly initialized and retrained models)?

Line 122 “We expect to achieve more robust unlearning that better withstands such post-unlearning attacks.” If FADE measures how well unlearned models mimic a retained model, why would FADE help make unlearning methods more robust to post-unlearning attacks? My understanding is that the latent information remains inside the model and it is not revealed at the logits/output until after the attack.

What is the variance of FADE across multiple unlearned models compared to the same reference model?

One could argue that it shouldn’t matter if an unlearned model maps the target concept to reasonable concept A vs reasonable concept B, but this could significantly impact FADE. What is the variance of FADE in this case, and should these semantics matter in practical scenarios?

---

> ### Author Response · Authors · 2025-11-20
> **Official Response to Reviewer 1R1j (1/2)**
>
> We thank Reviewer 1R1j for the constructive comments and suggestions. Below are our responses to your questions on our work.
>
> ---
> > [W1] Wouldn’t it be more appropriate to measure the expected FADE over distributions of models instead of distributions of outputs?
>
> &rarr; We thank the reviewer for sharing the reference [A] and insightful observation. As Definition 2 of [A] suggests, exact unlearning can be interpreted in two notions: one based on distributions over model weights and another based on distributions over model outputs. **Both are valid definitions.** However, as evaluating weight distributions would be computationally intractable for LLMs and diffusion models, FADE adheres to the latter and measures exact unlearning in the output space, serving as a theoretically principled and practically viable alternative.
>
> [A] [Nguyen et al., A Survey of Machine Unlearning. ACM Journal 2025.](https://dl.acm.org/doi/10.1145/3749987)
>
> ---
> > [W2,Q2,Q4] Is FADE stable when you test it with different randomly initialized and retrained models?
>
> &rarr; We thank the reviewer for the question. To assess FADE's stability under stochasticity, we conducted additional TOFU experiments computing FADE between unlearned and retain-only models using: (1) matched random seeds (our default setup) and (2) different seeds. All unlearning methods use LoRA rank 32 for 5 epochs.
>
> Results show FADE values remain largely stable across two setups for most unlearning methods, with the exception of preference optimization-based approaches (i.e., NPO, DPO), which exhibit higher variance. This suggests these methods are more sensitive to randomness in batch ordering or weight initialization. Nonetheless, **all methods consistently produce FADE values far from the retain-only baseline regardless of choice in seeding**, confirming our main finding is robust to difference in randomizations. For accurate evaluation, **we recommend using matched seeding as the retain-only model can serve as the true counterfactual had the forget data not been introduced during training**.
>
>
> | TOFU-1% | Matched seeds (ours) | Different seeds |
> |---:|:---:|:---:|
> | Base | $33.1 \pm 0.7$ | $33.1 \pm 0.7$ |
> | GA | $51.6 \pm 0.1$ | $51.5 \pm 1.6$ |
> | GD | $54.4 \pm 1.5$ | $52.7 \pm 1.8$ |
> | NPO | $50.9 \pm 0.3$ | $51.3 \pm 1.6$ |
> | DPO | $50.9 \pm 1.3$ | $49.6 \pm 1.9$ |
> | IHL | $51.6 \pm 1.8$ | $51.6 \pm 1.8$ |
> | Retain | $12.6 \pm 0.4$ | $12.6 \pm 0.4$ |
>
> | TOFU-5% | Matched seeds (ours) | Different seeds |
> |---:|:---:|:---:|
> | Base | $31.8 \pm 0.5$ | $31.8 \pm 0.5$ |
> | GA | $1013.4 \pm 313.2$ | $992.3 \pm 361.3$ |
> | GD | $144.3 \pm 44.1$ | $148.1 \pm 36.6$ |
> | NPO | $145.4 \pm 58.3$ | $75.9 \pm 4.9$ |
> | DPO | $39.6 \pm 1.5$ | $53.0 \pm 5.6$ |
> | IHL | $61.9 \pm 6.9$ | $61.8 \pm 6.6$ |
> | Retain | $10.9 \pm 0.6$ | $10.9 \pm 0.6$ |
>
> | TOFU-10% | Matched seeds (ours) | Different seeds |
> |---:|:---:|:---:|
> | Base | $32.1 \pm 0.4$ | $32.1 \pm 0.4$ |
> | GA | $3151.9 \pm 926.7$ | $2538.6 \pm 249.5$ |
> | GD | $3181.1 \pm 2225.2$ | $2612.8 \pm 1397.4$ |
> | NPO | $155.0 \pm 8.3$ | $101.9 \pm 65.0$ |
> | DPO | $49.0 \pm 1.6$ | $93.3 \pm 64.0$ |
> | IHL | $373.0 \pm 96.8$ | $373.0 \pm 97.6$ |
> | Retain | $10.5 \pm 0.2$ | $10.5 \pm 0.2$ |
>
>
> ---
> > [Q1] What is “functional alignment”?
>
> &rarr; We consider "functional alignment" to mean the resemblance between two models in their input-to-output mapping behavior. For stochastic models such as LLMs and T2I diffusion models, this means **the distribution of possible outputs given a prompt should be similar**. This differs from reference-based metrics, which compare against a single output rather than output distributions. We will add this clarification in the Introduction section.
>
> ---
> > [Q3] If FADE measures how well unlearned models mimic a retained model, why would FADE help make unlearning methods more robust to post-unlearning attacks?
>
> &rarr; We thank the reviewer for this insightful observation. We agree that output-level indistinguishability does not guarantee robustness to all post-unlearning attacks, as attacks may exploit latent representations not fully reflected in output distributions. However, we argue that **achieving low FADE is a necessary (though not sufficient) condition for robustness to post-unlearning attacks**: if a model's output distribution is distinguishable from the retain-only baseline, it certainly has not achieved unlearning. Our findings demonstrate that existing methods fail even this necessary first step. We will revise the Related Work section to clarify that FADE measures a prerequisite of post-unlearning robustness rather than a guarantee.

---

> ### Author Response · Authors · 2025-11-20
> **Official Response to Reviewer 1R1j (2/2)**
>
> ---
> > [Q5] Does it matter if an unlearned model maps the target concept to a reasonable concept different from the concept shown by the retain model?
>
> &rarr; We agree that achieving any "reasonable" concept should be rewarded: this is precisely how FADE is designed. **Retain-only models obtained from different seeds define the range of acceptable outputs, and FADE measures how closely the unlearned models can match this output distribution**.
>
> However, our qualitative results (Table 1 for TOFU, Figure 6 for UnlearnCanvas) reveal two key findings: (1) **the range of "reasonable" outputs representing true lack of learning is fairly specific**, as retain-only LLMs consistently assign high probability to the same semantically appropriate answers and retain-only T2I diffusion models exhibit consistent image styling; (2) **unlearned models generate outputs that consistently fall outside this range**, producing outputs that FADE correctly identifies as deviating from the distribution of retain-models, unlike previous metrics that fail to capture this distinction.

---

### Official Review · Reviewer_7Aru · 2025-10-31

**Soundness:** 3
**Presentation:** 2
**Contribution:** 1
**Rating:** 4
**Confidence:** 3

**Summary:**

This paper proposes a distribution-level unlearning metric FADE, to address the limitations of previous reference-specific approaches. Moreover, its experiments expose the failure of existing unlearning methods under this new metric.

**Strengths:**

1. This paper effectively reveals the problems that exist in current evaluation metrics.
2. The experiments are comprehensive.

**Weaknesses:**

1. This paper points out many shortcomings in current evaluation methods. This part is convincing but not surprising. After that, the paper proposes a new method called FADE. The main conclusion from the experiments is that under the FADE metric, existing unlearning methods perform poorly. However, the paper does not go on to propose more effective unlearning methods, making it feel quite incomplete. Based on the presented results, I cannot confirm that FADE is a flawless metric that could be widely accepted. Overall, the paper gives me the impression of lacking significant conclusions and constructive insight.
2. The LLM and T2I parts do not feel like a cohesive whole. Moreover, the FADE mentioned in line 245 and the formula in line 266 appear to be two completely different metrics, yet they share the same name. The discussions of these two parts are also quite disconnected, making the paper difficult to read.
3. From Figure 4, I'm not convinced by the author's claim in Line 228 that unlearned models generate inconsistent images. I think the three images from Ediff/ ESD share a similar style.

**Questions:**

I notice that retrained models are finetuned for 5 epochs on the retained dataset, but the unlearned models are tuned with LoRA from the base model.   I'm wondering if LoRA fine-tuning itself introduced some stuff you missed. To verify it, you can continue to tune a retained model with LoRA for 5 epochs and denote the resulting models as LoRA-Retain models. Then you can get a new baseline like the dashed line in Figure 5 by averaging over randomness.

---

> ### Author Response · Authors · 2025-11-20
> **Official Response to Reviewer 7Aru (1/1)**
>
> We thank Reviewer 7Aru for the constructive comments and suggestions. Below are our responses to your questions on our work.
>
> ---
> > [W1] Shortcomings in current evaluation methods are convincing but not surprising... the paper does not go on to propose more effective unlearning methods, making it incomplete.
>
> &rarr; While identifying flaws in existing metrics may appear unsurprising, the implications are significant: The Forget Quality metric used in TOFU as well as style-classifier-based metrics in UnlearnCanvas have been adopted as standard by numerous papers, meaning **substantial recent work have been optimizing toward fundamentally misaligned objectives**. Revealing this misalignment is a critical insight that redirects the field.
>
> Regarding the absence of a proposed method for unlearning: we believe **establishing a correct evaluation framework is a contribution that can stand independently**. By establishing FADE independently, we propose a principled foundation upon which the community can develop and fairly compare future methods.
>
> ---
> > [W2] The LLM and T2I parts do not feel like a cohesive whole.
>
> &rarr; We thank the reviewer for this feedback. We would like to clarify that despite the apparent differences in formulation, they are only computational adaptations reflecting the different generative procedures (autoregressive vs. denoising). **Both instantiations follow the same probabilistic concept of FADE**, which is to measure the symmetrized divergence between unlearned and retain model output distributions (See Lines 250-294 of the manuscript).
>
> We intentionally cover both modalities to demonstrate that (1) **reference-based evaluation is a cross-domain problem** affecting both LLM and T2I unlearning research, and (2) **FADE provides a modality-agnostic solution** based on a unified probabilistic principle. This breadth strengthens our contribution by showing the generality of our findings. We will clarify this unified foundation in the introduction section for better readability.
>
> ---
> > [W3] On Figure 4: I'm not convinced by the author's claim in Line 228 that unlearned models generate inconsistent images. I think the three images from Ediff/ESD share a similar style.
>
> &rarr; The key finding of Figure 4 is that **unlearned methods (e.g., EDiff, ESD) are unable to replicate the style shown by the retain-only model** (which is the gold-standard for T2I unlearning), not that EDiff and ESD each exhibit consistent styles within itself. We will revise Lines 216-224 to better clarify this point.
>
> ---
> > [Q1] Why not tune a retained model with LoRA for 5 epochs and use the resulting models as another baseline called LoRA-Retain?
>
> &rarr; We thank the reviewer for this suggestion. However, we respectfully clarify that **the only gold-standard baseline in our TOFU experiments is the full-finetuned retain-only model**. This model must be trained using the identical optimization process as the original base model, but with the forget data removed from the training set. This ensures the retain-only model represents the model that would have been obtained if the forget data had never been included in the first place.
>
> Following previous work, **the base model in our TOFU experiments is obtained through full fine-tuning (not LoRA), so the retain-only baseline must also use full fine-tuning** to maintain consistency with this training pipeline. The unlearning methods using LoRA are post-hoc interventions on the already-trained base model, and thus their optimization process is independent from that of retain-only models. Comparing LoRA-based retain-only training to LoRA-based unlearning would not reflect the true unlearning scenario, where the goal is to approximate retraining from scratch on the retain set.

---

### Official Review · Reviewer_8y78 · 2025-10-31

**Soundness:** 3
**Presentation:** 3
**Contribution:** 2
**Rating:** 4
**Confidence:** 3

**Summary:**

The paper examines current unlearning evaluation metrics for generative models, arguing that prevalent reference-specific metrics fail to assess true unlearning. The paper introduces FADE to quantify distributional similarity between an unlearned model and a retain-only model using a bidirectional likelihood comparison on generated outputs. Experiments on LLM unlearning and T2I demonstrate that unlearning methods may appear successful under traditional metrics but exhibit significant distributional discrepancies not captured by those evaluations, whereas FADE exposes these failures.

**Strengths:**

1. The paper identifies and convincingly demonstrates the limitations of current unlearning evaluation practices, providing empirical and theoretical support for the claim that reference-specific metrics could result in overestimating unlearning efficacy.
2. The problem and motivation identified in the paper are novel. FADE is implemented in a way that is modality-agnostic, working for both autoregressive language models and diffusion models.
3. The work addresses a significant and growing issue as unlearning gains attention for safety/privacy/ethical AI deployment. The results challenge prevailing practices and set a high standard for future work in the area.

**Weaknesses:**

1.  The mathematical description of FADE is clear. Still, the paper does not seem to present deeper theoretical guarantees or formal links between FADE and true indistinguishability in the full probabilistic sense.
2. The core of the proposed method lies in using comparative scores to evaluate the differences between the unlearned model and the retain-only model. That said, it seems that one potentially informative control experiment is absent- Namely, assessing the comparative scores between different unlearned models. Incorporating such an experiment could help enhance the generality of the findings and strengthen the rigor of the validation.
3. It appears that the paper does not include a discussion or experimental evaluation of the computational/time/token cost associated with the newly proposed metric, which could be an important aspect. Addressing this point may provide a more comprehensive assessment of the metric’s practicality.
4. Although the writing is generally clear and easy to follow, there appear to be numerous instances that resemble AI-generated text patterns (the frequent use of “–” symbols), though I may be mistaken.
5. In practice, likelihood-based comparison metrics (such as FADE) can be sensitive to the sampling strategy. The paper does not explore whether alternative sampling schemes (diverse decoding, hard negative mining, etc.) could strengthen or weaken FADE as a metric.
6. there’s comparatively little discussion of hyperparameter, optimizer, or architecture robustness

**Questions:**

1. Could the authors provide comparative experimental results regarding computational cost and time overhead?
2. Would it be possible for the authors to include control experiment results involving mutual scoring between two unlearned models?

---

> ### Author Response · Authors · 2025-11-20
> **Official Response to Reviewer 8y78 (1/3)**
>
> We thank Reviewer 8y78 for the constructive comments and suggestions. Below are our responses to your questions on our work.
>
> ---
> > [W1] Any theoretical guarantees or formal links between FADE and true indistinguishability in the full probabilistic sense?
>
> &rarr; Thank you for the insightful question. Based on the fact that FADE is a Monte-Carlo estimate of the bidirectional KL-divergence between two output distributions, we can draw a connection between FADE and the total variation distance. As Pinsker's inequality states the total variation distance $D_{\text{TV}}$ between the two distributions is upper-bounded by the KL-divergence $D_{\text{KL}}$, FADE effectively upper-bounds the TV-distance, leading to the following inequality:
> $$
> D^2_{\text{TV}}(P_{\text{retain}}, P_{\text{unlearned}}) \leq \dfrac{1}{4}[D_{\text{KL}}(P_{\text{retain}}, P_{\text{unlearned}})+ D_{\text{KL}}(P_{\text{unlearned}}, P_{\text{retain}})] \approx \dfrac{1}{2}\text{FADE}
> $$
> This inequality suggests that **a smaller FADE implies a smaller total variation distance, which directly implies the output distributions are harder to distinguish via any statistical test**. This connection also opens interesting research questions: (1) How does TV-distance relate to notions of indistinguishability in unlearning (e.g., differential privacy)?, and (2) what is the convergence behavior of FADE in estimating the KL-divergence? We view these as potential directions for future work and will add this discussion to the Method section.
>
> ---
> > [W2,Q2] Missing a potentially informative control experiment: comparing scores between different unlearned models.
>
> &rarr; We thank the reviewer for the suggestion. We measured FADE between pairs of unlearned models using different methods, revealing interesting patterns. The results below show that (1) for small forget sets (TOFU-1%), different methods exhibit similar output distributions, indicating their output distributions deviate similarly from that of the base model, and (2) for larger forget sets (TOFU-5% and 10%), methods diverge from each other significantly with high variance. Together with our main results in Figure 5, these results imply that **existing unlearning methods lack a mechanism to approach the correct solution and instead deviate from the retain-only baseline in directions specific to each method**. We will add these results to the Appendix.
>
>
> | Method 1 | Method 2 | TOFU-1% | TOFU-5% | TOFU-10%|
> |:---:|:---:|:---:|:---:|:---:|
> | GA | GD | $0.6 \pm 0.3$ | $2387.1 \pm 1693.8$ | $19021.8 \pm 8768.8$ |
> | GA | NPO | $0.2 \pm 0.0$ | $2151.9 \pm 1103.0$ | $12146.6 \pm 919.9$ |
> | GA | DPO | $2.9 \pm 0.3$ | $581.3 \pm 105.7$ | $2318.1 \pm 875.5$ |
> | GA | IHL | $0.3 \pm 0.1$ | $1628.8 \pm 629.1$ | $10246.0 \pm 5485.6$ |
> | GD | NPO | $0.7 \pm 0.3$ | $434.0 \pm 196.6$ | $11740.5 \pm 7431.0$ |
> | GD | DPO | $3.4 \pm 0.5$ | $97.2 \pm 35.9$ | $2298.2 \pm 1734.6$ |
> | GD | IHL | $0.4 \pm 0.1$ | $161.7 \pm 100.7$ | $6276.7 \pm 1256.5$ |
> | NPO | DPO | $2.7 \pm 0.2$ | $92.8 \pm 35.1$ | $140.5 \pm 6.9$ |
> | NPO | IHL | $0.4 \pm 0.1$ | $104.7 \pm 37.3$ | $1182.1 \pm 315.9$ |
> | DPO | IHL | $2.9 \pm 0.5$ | $34.6 \pm 5.1$ | $258.6 \pm 77.5$ |
>
> ---
> > [W3,Q1] Missing a discussion of the computational cost associated with FADE, which could be an important aspect.
>
> &rarr; We acknowledge that FADE computation requires non-trivial cost. For reference, the table below shows the runtime of computing FADE with 100 samples per prompt between the base and retain-only models on TOFU tasks requires on A100 GPUs. However, we emphasize three points:
>
> First, **evaluation accuracy far outweighs the cost in compute**. Rigorous evaluation is a necessity for directing research toward correct objectives, yet cost-efficient but inaccurate metrics have been misdirecting substantial work: a far greater cost than compute for proper evaluation.
>
> Second, **FADE is highly parallelizable**. While we conducted experiments on a single GPU, both steps of computing FADE (sampling and likelihood estimation) can be distributed across multiple GPUs, then aggregated for the final estimation, significantly reducing runtime in practice.
>
> Third, **the cost can be reduced with minimal impact on our findings**. Our ablation study (please refer to our response to [Q6] of Reviewer 8y78) shows that FADE remains robust with as few as 10 samples per prompt, allowing practitioners to adjust compute cost while preserving key trends.
>
> | Runtime (in minutes) | TOFU-1% | TOFU-5% | TOFU-10%|
> |:---|:---:|:---:|:---:|
> | Monte-Carlo Sampling | $2.0 \pm 0.2$ | $12.1 \pm 1.3$ | $17.2 \pm 2.4$ |
> | Likelihood Estimation | $0.4 \pm 0.1$ | $2.7 \pm 0.1$ | $5.6 \pm 0.1$ |
>
> ---
> > [W4] Frequent use of “–” symbols resembling AI-generated text.
>
> &rarr; Our use of "-\-\-" symbols was intended to emphasize phrases in the paper. Nonetheless, we agree that such use can be misleading and distracting: we will revise the writing.

---

> ### Author Response · Authors · 2025-11-20
> **Official Response to Reviewer 8y78 (2/3)**
>
> ---
> > [W5] Likelihood-based comparison metrics such as FADE can be sensitive to the sampling strategy.
>
> &rarr; We thank the reviewer for the question on the robustness of FADE to different decoding strategies. Our FADE computation for LLMs uses default multinomial sampling, ensuring unbiased likelihood divergence estimation. However, to address the reviewer's concern, we conducted additional experiments using various sampling strategies: **Nucleus Sampling (NS) with parameter $p \in \\{0.95, 0.9, 0.7, 0.5\\}$ and Beam Search (BS) with beam sizes $B \in \\{25, 50, 100\\}$**. Note that these sampling-based variants may introduce bias in the divergence estimation compared to our default approach.
>
> The results shown below from TOFU after 5 epochs of unlearning with each unlearning method suggests that (1) **FADE values remain consistent across different decoding strategies**, and (2) our core finding holds: **regardless of decoding strategy, existing methods fail to approach the retain-only baseline**. This confirms that **our main findings are not a result of a particular choice in decoding strategy**. We will add these ablation results to the appendix.
>
> | TOFU-1% | Default (ours) | NS $(p=0.95)$ | NS $(p=0.9)$ | NS $(p=0.7)$ | NS $(p=0.5)$ | BS $(B=25)$ | BS $(B=50)$ | BS $(B=100)$ |
> |---:|:---:|:---:|:---:|:---:|:---:|:---:|:---:|:---:|
> | Base | $33.1 \pm 0.7$ | $30.7 \pm 1.1$ | $30.8 \pm 0.9$ | $30.5 \pm 1.4$ | $30.6 \pm 1.1$ | $30.8 \pm 0.9$ | $31.1 \pm 1.2$ | $31.3 \pm 1.3$ |
> | GA | $51.6 \pm 0.1$ | $52.1 \pm 0.7$ | $52.2 \pm 0.8$ | $52.5 \pm 1.2$ | $52.7 \pm 1.1$ | $49.4 \pm 0.8$ | $47.6 \pm 0.8$ | $45.6 \pm 0.7$ |
> | GD | $54.4 \pm 1.5$ | $54.3 \pm 1.5$ | $54.7 \pm 1.6$ | $54.8 \pm 1.7$ | $55.0 \pm 1.9$ | $50.2 \pm 0.9$ | $48.1 \pm 1.3$ | $46.4 \pm 1.2$ |
> | NPO | $50.9 \pm 0.3$ | $51.4 \pm 0.9$ | $51.7 \pm 0.76$ | $51.7 \pm 1.5$ | $52.5 \pm 1.3$ | $49.1 \pm 0.9$ | $47.6 \pm 0.8$ | $45.5 \pm 0.8$ |
> | DPO | $50.9 \pm 1.3$ | $50.8 \pm 1.5$ | $51.1 \pm 1.7$  | $51.2 \pm 2.1$ | $52.6 \pm 2.7$ | $47.3 \pm 1.3$ | $45.5 \pm 1.4$ | $43.2 \pm 1.4$ |
> | IHL | $51.6 \pm 1.8$ | $51.9 \pm 1.5$ | $52.0 \pm 1.4$ | $52.6 \pm 1.6$ | $54.0 \pm 0.8$ | $49.0 \pm 1.4$ | $47.3 \pm 1.1$ | $45.6 \pm 1.4$ |
> | Retain | $12.6 \pm 0.4$ | $10.3 \pm 1.1$ | $10.2 \pm 1.0$ | $10.2 \pm 1.0$ | $10.4 \pm 1.4$ | $9.8 \pm 0.5$ | $10.1 \pm 0.3$ | $10.3 \pm 0.4$ |
>
> | TOFU-5% | Default (ours) | NS $(p=0.95)$ | NS $(p=0.9)$ | NS $(p=0.7)$ | NS $(p=0.5)$ | BS $(B=25)$ | BS $(B=50)$ | BS $(B=100)$ |
> |---:|:---:|:---:|:---:|:---:|:---:|:---:|:---:|:---:|
> | Base | $31.8 \pm 0.5$ | $30.1 \pm 1.2$ | $30.0 \pm 1.1$ | $30.3 \pm 1.2$ | $30.1 \pm 1.3$ | $29.2 \pm 0.9$ | $29.4 \pm 1.0$ | $29.6 \pm 0.9$ |
> | GA | $1013.4 \pm 313.2$ | $933.3 \pm 299.8$ | $929.0 \pm 297.6$ | $928.4 \pm 296.9$ | $932.6 \pm 305.6$ | $936.5 \pm 290.6$ | $935.6 \pm 289.8$ | $932.2 \pm 286.9$ |
> | GD | $144.3 \pm 44.1$ | $139.2 \pm 43.2$ | $138.1 \pm 42.4$ | $137.6 \pm 42.3$ | $137.1 \pm 42.9$ | $126.8 \pm 36.7$ | $126.5 \pm 36.5$ | $127.4 \pm 36.7$ |
> | NPO | $145.4 \pm 58.3$ | $137.0 \pm 53.6$ | $136.4 \pm 53.1$ | $134.4 \pm 51.7$ | $132.9 \pm 51.1$ | $124.2 \pm 44.0$ | $122.2 \pm 42.3$ | $120.2 \pm 40.8$ |
> | DPO | $39.6 \pm 1.5$ | $36.9 \pm 1.4$ | $36.9 \pm 1.3$ | $37.1 \pm 1.5$ | $36.9 \pm 1.4$ | $36.7 \pm 1.5$ | $35.3 \pm 1.3$ | $34.3 \pm 1.1$ |
> | IHL | $61.9 \pm 6.9$ | $57.9 \pm 6.4$ | $57.8 \pm 6.4$ | $57.6 \pm 6.2$ | $57.8 \pm 6.4$ | $51.9 \pm 4.6$ | $51.3 \pm 4.2$ | $50.9 \pm 4.2$ |
> | Retain | $10.9 \pm 0.6$ | $9.6 \pm 1.0$ | $9.5 \pm 0.9$ | $9.6 \pm 0.9$ | $9.5 \pm 0.9$ | $9.0 \pm 0.9$ | $9.1 \pm 1.0$ | $9.3 \pm 1.0$ |
>
> | TOFU-10% | Default (ours) | NS $(p=0.95)$ | NS $(p=0.9)$ | NS $(p=0.7)$ | NS $(p=0.5)$ | BS $(B=25)$ | BS $(B=50)$ | BS $(B=100)$ |
> |---:|:---:|:---:|:---:|:---:|:---:|:---:|:---:|:---:|
> | Base | $32.1 \pm 0.4$ | $29.7 \pm 1.1$ | $29.6 \pm 1.0$ | $29.7 \pm 1.1$ | $29.8 \pm 1.2$ | $28.8 \pm 0.4$ | $29.0 \pm 0.3$ | $29.1 \pm 0.3$ |
> | GA | $3151.9 \pm 926.7$ | $2924.8 \pm 862.0$ | $2924.2 \pm 861.7$ | $2925.1 \pm 862.7$ | $2935.4 \pm 849.7$ | $3012.7 \pm 992.5$ | $3014.5 \pm 991.2$ | $3005.1 \pm 988.0$ |
> | GD | $3181.1 \pm 2225.2$ | $2955.9 \pm 2164.0$ | $2956.0 \pm 2164.5$ | $2956.3 \pm 2164.9$ | $2968.1 \pm 2174.6$ | $2460.6 \pm 1374.8$ | $2436.1 \pm 1337.6$ | $2397.3 \pm 1292.7$ |
> | NPO | $155.0 \pm 8.3$ | $135.7 \pm 12.3$ | $133.6 \pm 12.7$ | $128.7 \pm 13.5$ | $126.5 \pm 14.0$ | $116.6 \pm 20.3$ | $112.9 \pm 19.6$ | $109.2 \pm 18.9$ |
> | DPO | $49.0 \pm 1.6$ | $46.8 \pm 1.7$ | $46.9 \pm 1.7$ | $47.1 \pm 1.8$ | $47.0 \pm 1.6$ | $42.8 \pm 1.5$ | $41.5 \pm 1.3$ | $40.6 \pm 1.1$ |
> | IHL | $373.0 \pm 96.8$ | $341.4 \pm 84.1$ | $341.5 \pm 84.1$ | $341.3 \pm 83.6$ | $342.3 \pm 83.0$ | $353.6 \pm 90.4$ | $353.8 \pm 90.0$ | $352.1 \pm 89.5$ |
> | Retain | $10.5 \pm 0.2$ | $9.1 \pm 0.5$ | $9.0 \pm 0.5$ | $9.0 \pm 0.5$ | $8.9 \pm 0.7$ | $8.4 \pm 0.3$ | $8.6 \pm 0.3$ | $8.7 \pm 0.3$ |

---

> ### Author Response · Authors · 2025-11-20
> **Official Response to Reviewer 8y78 (3/3)**
>
> ---
> > [W6] Discussion of hyperparameter, optimizer, or architecture robustness.
>
> &rarr; All hyperparameters, optimizers, and architectures follow the standard settings from the TOFU [A] and UnlearnCanvas [B] benchmarks to ensure fair comparison with existing work. The only hyperparameter additionally introduced in our work is the number of samples $N$ generated per prompt for computing FADE.
>
> To assess robustness, we conducted ablation experiments on the TOFU benchmark with varying sample sizes $N \in \\{100, 50, 25, 10\\}$, all unlearning methods using LoRA with rank 32 for 5 epochs. The results shown below demonstrate that FADE values remain consistent across different sample sizes, and more critically, our main finding holds universally: **all unlearning methods fail to reduce FADE toward the retain-only baseline regardless of $N$**. We will add these robustness results to the Experimental Results section.
>
> | TOFU-1% | N=100 | 50 | 25 | 10 |
> |---:|:---:|:---:|:---:|:---:|
> | Base | $33.1 \pm 0.7$ | $33.1 \pm 0.5$ | $33.3 \pm 0.8$ | $33.0 \pm 1.2$ |
> | GA | $51.6 \pm 0.1$ | $51.6 \pm 0.3$ | $51.2 \pm 0.1$ | $51.2 \pm 0.5$ |
> | GD | $54.4 \pm 1.5$ | $54.3 \pm 1.3$ | $54.3 \pm 1.5$ | $54.2 \pm 0.7$ |
> | NPO | $50.9 \pm 0.3$ | $50.6 \pm 0.4$ | $50.8 \pm 0.4$ | $50.8 \pm 1.3$ |
> | DPO | $50.9 \pm 1.3$ | $50.9 \pm 1.3$ | $51.0 \pm 1.2$ | $50.3 \pm 1.5$ |
> | IHL | $51.6 \pm 1.8$ | $51.4 \pm 1.9$ | $51.6 \pm 1.7$ | $52.0 \pm 2.3$ |
> | Retain | $12.6 \pm 0.4$ | $12.6 \pm 0.3$ | $12.8 \pm 0.4$ | $12.7 \pm 0.6$ |
>
> | TOFU-5% | N=100 | 50 | 25 | 10 |
> |---:|:---:|:---:|:---:|:---:|
> | Base | $31.8 \pm 0.5$ | $31.7 \pm 0.5$ | $31.7 \pm 0.5$ | $31.9 \pm 0.4$ |
> | GA | $1013.4 \pm 313.2$ | $1013.6 \pm 314.7$ | $1015.4 \pm 311.6$ | $1016.3 \pm 314.3$ |
> | GD | $144.3 \pm 44.1$ | $144.5 \pm 44.1$ | $145.1 \pm 44.6$ | $144.7 \pm 44.1$ |
> | NPO | $145.4 \pm 58.3$ | $145.2 \pm 58.2$ | $145.7 \pm 58.4$ | $145.4 \pm 58.7$ |
> | DPO | $39.6 \pm 1.5$ | $39.5 \pm 1.5$ | $39.5 \pm 1.6$ | $39.5 \pm 1.3$ |
> | IHL | $61.9 \pm 6.9$ | $62.1 \pm 7.1$ | $61.8 \pm 7.1$ | $61.9 \pm 7.0$ |
> | Retain | $10.9 \pm 0.6$ | $10.8 \pm 0.5$ | $10.8 \pm 0.5$ | $10.8 \pm 0.6$ |
>
> | TOFU-10% | N=100 | 50 | 25 | 10 |
> |---:|:---:|:---:|:---:|:---:|
> | Base | $32.1 \pm 0.4$ | $32.1 \pm 0.4$ | $32.2 \pm 0.4$ | $32.0 \pm 0.4$ |
> | GA | $3151.9 \pm 926.7$ | $3152.5 \pm 926.2$ | $3152.5 \pm 926.8$ | $3157.9 \pm 929.7$ |
> | GD | $3181.1 \pm 2225.2$ | $3181.7 \pm 2227.0$ | $3178.5 \pm 2219.3$ | $3185.1 \pm 2226.4$ |
> | NPO | $155.0 \pm 8.3$ | $155.1 \pm 8.0$ | $155.2 \pm 8.5$ | $155.2 \pm 8.1$ |
> | DPO | $49.0 \pm 1.6$ | $49.0 \pm 1.5$ | $49.0 \pm 1.6$ | $49.1 \pm 1.5$ |
> | IHL | $373.0 \pm 96.8$ | $373.3 \pm 96.9$ | $373.4 \pm 97.1$ | $373.4 \pm 97.7$ |
> | Retain | $10.5 \pm 0.2$ | $10.4 \pm 0.2$ | $10.4 \pm 0.1$ | $10.5 \pm 0.4$ |
>
> [A] [Maini et al., TOFU: A Task of Fictitious Unlearning for LLMs. COLM 2024.](https://arxiv.org/abs/2401.06121)\
> [B] [Zhang et al., UnlearnCanvas: Stylized Image Dataset for Enhanced Machine Unlearning Evaluation in Diffusion Models. NeurIPS D&B Track 2024.](https://arxiv.org/abs/2402.11846)

---

### Official Review · Reviewer_PTzk · 2025-11-01

**Soundness:** 3
**Presentation:** 3
**Contribution:** 3
**Rating:** 6
**Confidence:** 4

**Summary:**

This paper highlights an existing problem in unlearning: unreliable metrics for verifying unlearning for generative models. The paper proposes Functional Alignment for Distributional Equivalence (FADE), which evaluates distributional alignment with retain-only oracles through bidirectional likelihood comparisons over generated samples. The experimental results evaluate the FADE score on language models and text-to-image diffusion models and several well-known unlearning methods. It shows that existing metrics such as unlearning accuracy can be misleading for evaluating the efficacy of unlearning.

**Strengths:**

1. Grounded definition. The paper frames unlearning as achieving functional alignment with a retain-only model (i.e., behaving as if the forgotten data was never seen). This aligns with the gold standard (exact perfect unlearning) and provides a clear conceptual foundation.



2. Points to a significant gap in existing work. The authors highlight that current evaluation methods rely on reference-specific proxies (e.g., fixed answers, classifiers), which can mask failures and even allow recovery attacks. Thus, the work exposes significant blind spots in prevailing practice.



3. FADE measures full distributional equivalence bidirectionally, not just task-specific correctness. It applies across modalities (LLMs and diffusion models) and detects subtle failures that reference-conditioned metrics overlook.

**Weaknesses:**

1. Application. While the FADE metric and evaluation are important to emphasize existing problems, FADE requires having a retain-only model. Thus, it is not clear whether it can be practically implemented (or a proxy of it) to improve the unlearning method itself.

2. Computationally expensive. It is computationally expensive to compute the retain-only model under different seeds since it requires training the model without the forget set samples. FADE requires Monte-Carlo style sampling and likelihood estimates (or denoising approximations for diffusion models), which may be expensive and subject to variance depending on chosen sampling strategies.

**Questions:**

1. What do we learn from the FADE metric and evaluation for future unlearning methods? It would be good if the authors can comment on how they see their metric applied in evaluations and future unlearning methods.

2. How sensitive is FADE to generation strategy? Since LLM measurement relies on top-k/nucleus sampling, do different decoding strategies change FADE outcomes? How consistent are evaluations across sampling choices?

---

> ### Author Response · Authors · 2025-11-20
> **Official Response to Reviewer PTzk (1/2)**
>
> We thank Reviewer PTzk for the constructive comments and suggestions. Below are our responses to your questions on our work.
>
> ---
> > [W1] FADE requires having a retain-only model that is computationally expensive to obtain. Thus, it is not clear whether it can be practically implemented.
>
> &rarr; Thank you for raising this practical consideration. We address two key points:
>
> First, **FADE is designed as a gold-standard evaluation metric for selecting effective unlearning algorithms, not as a requirement during real-world deployment**. While obtaining retain-only models does incur additional compute, this is only paid once during evaluation. After an unlearning method is validated using FADE, the method itself can be deployed in practical settings without needing any retain-only model.
>
> Second, **this requirement is already under standard practice in unlearning research: the Forget Quality (FQ) metric used in the TOFU benchmark also requires retain-only models** (see Lines 151-153 and [A]). The wide use of the benchmark clearly demonstrates that this requirement is not a barrier to practical research. Following TOFU, we will also fully open-source our base and retain-only model checkpoints alongside our codebase upon acceptance to facilitate efficient future research.
>
> [A] [Maini et al., TOFU: A Task of Fictitious Unlearning for LLMs. COLM 2024.](https://arxiv.org/abs/2401.06121)
>
> ---
> > [W2] FADE also requires Monte-Carlo style sampling and likelihood estimates which may be expensive.
>
> &rarr; We acknowledge that FADE computation requires non-trivial cost. For reference, the table below shows the runtime of computing FADE with 100 samples per prompt between the base and retain-only models on TOFU tasks requires on A100 GPUs. However, we emphasize three points:
>
> First, **evaluation accuracy far outweighs the cost in compute**. Rigorous evaluation is a necessity for directing research toward correct objectives, yet cost-efficient but inaccurate metrics have been misdirecting substantial work: a far greater cost than compute for proper evaluation.
>
> Second, **FADE is highly parallelizable**. While we conducted experiments on a single GPU, both steps of computing FADE (sampling and likelihood estimation) can be distributed across multiple GPUs, then aggregated for the final estimation, significantly reducing runtime in practice.
>
> Third, **the cost can be reduced with minimal impact on our findings**. Our ablation study (please refer to our response to [W6] of Reviewer 8y78) shows that FADE remains robust with as few as 10 samples per prompt, allowing practitioners to adjust compute cost while preserving the key trends.
>
> | Runtime (in minutes) | TOFU-1% | TOFU-5% | TOFU-10%|
> |:---|:---:|:---:|:---:|
> | Monte-Carlo Sampling | $2.0 \pm 0.2$ | $12.1 \pm 1.3$ | $17.2 \pm 2.4$ |
> | Likelihood Estimation | $0.4 \pm 0.1$ | $2.7 \pm 0.1$ | $5.6 \pm 0.1$ |
>
>
> ---
> > [Q1] What do we learn from the FADE metric and evaluation for future unlearning methods? How would FADE be applied in evaluations and future unlearning methods?
>
> &rarr; FADE serves as a measure of how close the unlearned model is to the gold-standard retain-only model with respect to its output distribution. As discussed in our response to [W1], FADE requires the same components as the Forget Quality (FQ) metric in TOFU, an unlearned model and a retain-only model, and thus **FADE can be applied in evaluations identically to how FQ in TOFU is currently used**. **The key difference is that FADE measures the correct objective**: indistinguishability in the output distribution space rather than on specific reference outputs.
>
> Once a new unlearning method is designed, researchers can compute FADE using the retain-only model checkpoints (which we will release) to quantify how closely the unlearned model matches the gold-standard. A lower FADE value will indicate more successful unlearning.

---

> ### Author Response · Authors · 2025-11-20
> **Official Response to Reviewer PTzk (2/2)**
>
> ---
> > [Q2] FADE can be subject to variance depending on chosen sampling strategies. How sensitive is FADE to generation strategy? Do different decoding strategies change FADE outcomes?
>
> &rarr; We thank the reviewer for the question on the robustness of FADE to different decoding strategies. Our FADE computation for LLMs uses default multinomial sampling, ensuring unbiased likelihood divergence estimation. However, to address the reviewer's concern, we conducted additional experiments using various sampling strategies: **Nucleus Sampling (NS) with parameter $p \in \\{0.95, 0.9, 0.7, 0.5\\}$ and Beam Search (BS) with beam sizes $B \in \\{25, 50, 100\\}$**. Note that these sampling-based variants may introduce bias in the divergence estimation compared to our default approach.
>
> The results shown below from TOFU after 5 epochs of unlearning with each unlearning method suggests that (1) **FADE values remain consistent across different decoding strategies**, and (2) our core finding holds: **regardless of decoding strategy, existing methods fail to approach the retain-only baseline**. This confirms that **our main findings are not a result of a particular choice in decoding strategy**. We will add these ablation results to the appendix.
>
> | TOFU-1% | Default (ours) | NS $(p=0.95)$ | NS $(p=0.9)$ | NS $(p=0.7)$ | NS $(p=0.5)$ | BS $(B=25)$ | BS $(B=50)$ | BS $(B=100)$ |
> |---:|:---:|:---:|:---:|:---:|:---:|:---:|:---:|:---:|
> | Base | $33.1 \pm 0.7$ | $30.7 \pm 1.1$ | $30.8 \pm 0.9$ | $30.5 \pm 1.4$ | $30.6 \pm 1.1$ | $30.8 \pm 0.9$ | $31.1 \pm 1.2$ | $31.3 \pm 1.3$ |
> | GA | $51.6 \pm 0.1$ | $52.1 \pm 0.7$ | $52.2 \pm 0.8$ | $52.5 \pm 1.2$ | $52.7 \pm 1.1$ | $49.4 \pm 0.8$ | $47.6 \pm 0.8$ | $45.6 \pm 0.7$ |
> | GD | $54.4 \pm 1.5$ | $54.3 \pm 1.5$ | $54.7 \pm 1.6$ | $54.8 \pm 1.7$ | $55.0 \pm 1.9$ | $50.2 \pm 0.9$ | $48.1 \pm 1.3$ | $46.4 \pm 1.2$ |
> | NPO | $50.9 \pm 0.3$ | $51.4 \pm 0.9$ | $51.7 \pm 0.76$ | $51.7 \pm 1.5$ | $52.5 \pm 1.3$ | $49.1 \pm 0.9$ | $47.6 \pm 0.8$ | $45.5 \pm 0.8$ |
> | DPO | $50.9 \pm 1.3$ | $50.8 \pm 1.5$ | $51.1 \pm 1.7$  | $51.2 \pm 2.1$ | $52.6 \pm 2.7$ | $47.3 \pm 1.3$ | $45.5 \pm 1.4$ | $43.2 \pm 1.4$ |
> | IHL | $51.6 \pm 1.8$ | $51.9 \pm 1.5$ | $52.0 \pm 1.4$ | $52.6 \pm 1.6$ | $54.0 \pm 0.8$ | $49.0 \pm 1.4$ | $47.3 \pm 1.1$ | $45.6 \pm 1.4$ |
> | Retain | $12.6 \pm 0.4$ | $10.3 \pm 1.1$ | $10.2 \pm 1.0$ | $10.2 \pm 1.0$ | $10.4 \pm 1.4$ | $9.8 \pm 0.5$ | $10.1 \pm 0.3$ | $10.3 \pm 0.4$ |
>
> | TOFU-5% | Default (ours) | NS $(p=0.95)$ | NS $(p=0.9)$ | NS $(p=0.7)$ | NS $(p=0.5)$ | BS $(B=25)$ | BS $(B=50)$ | BS $(B=100)$ |
> |---:|:---:|:---:|:---:|:---:|:---:|:---:|:---:|:---:|
> | Base | $31.8 \pm 0.5$ | $30.1 \pm 1.2$ | $30.0 \pm 1.1$ | $30.3 \pm 1.2$ | $30.1 \pm 1.3$ | $29.2 \pm 0.9$ | $29.4 \pm 1.0$ | $29.6 \pm 0.9$ |
> | GA | $1013.4 \pm 313.2$ | $933.3 \pm 299.8$ | $929.0 \pm 297.6$ | $928.4 \pm 296.9$ | $932.6 \pm 305.6$ | $936.5 \pm 290.6$ | $935.6 \pm 289.8$ | $932.2 \pm 286.9$ |
> | GD | $144.3 \pm 44.1$ | $139.2 \pm 43.2$ | $138.1 \pm 42.4$ | $137.6 \pm 42.3$ | $137.1 \pm 42.9$ | $126.8 \pm 36.7$ | $126.5 \pm 36.5$ | $127.4 \pm 36.7$ |
> | NPO | $145.4 \pm 58.3$ | $137.0 \pm 53.6$ | $136.4 \pm 53.1$ | $134.4 \pm 51.7$ | $132.9 \pm 51.1$ | $124.2 \pm 44.0$ | $122.2 \pm 42.3$ | $120.2 \pm 40.8$ |
> | DPO | $39.6 \pm 1.5$ | $36.9 \pm 1.4$ | $36.9 \pm 1.3$ | $37.1 \pm 1.5$ | $36.9 \pm 1.4$ | $36.7 \pm 1.5$ | $35.3 \pm 1.3$ | $34.3 \pm 1.1$ |
> | IHL | $61.9 \pm 6.9$ | $57.9 \pm 6.4$ | $57.8 \pm 6.4$ | $57.6 \pm 6.2$ | $57.8 \pm 6.4$ | $51.9 \pm 4.6$ | $51.3 \pm 4.2$ | $50.9 \pm 4.2$ |
> | Retain | $10.9 \pm 0.6$ | $9.6 \pm 1.0$ | $9.5 \pm 0.9$ | $9.6 \pm 0.9$ | $9.5 \pm 0.9$ | $9.0 \pm 0.9$ | $9.1 \pm 1.0$ | $9.3 \pm 1.0$ |
>
> | TOFU-10% | Default (ours) | NS $(p=0.95)$ | NS $(p=0.9)$ | NS $(p=0.7)$ | NS $(p=0.5)$ | BS $(B=25)$ | BS $(B=50)$ | BS $(B=100)$ |
> |---:|:---:|:---:|:---:|:---:|:---:|:---:|:---:|:---:|
> | Base | $32.1 \pm 0.4$ | $29.7 \pm 1.1$ | $29.6 \pm 1.0$ | $29.7 \pm 1.1$ | $29.8 \pm 1.2$ | $28.8 \pm 0.4$ | $29.0 \pm 0.3$ | $29.1 \pm 0.3$ |
> | GA | $3151.9 \pm 926.7$ | $2924.8 \pm 862.0$ | $2924.2 \pm 861.7$ | $2925.1 \pm 862.7$ | $2935.4 \pm 849.7$ | $3012.7 \pm 992.5$ | $3014.5 \pm 991.2$ | $3005.1 \pm 988.0$ |
> | GD | $3181.1 \pm 2225.2$ | $2955.9 \pm 2164.0$ | $2956.0 \pm 2164.5$ | $2956.3 \pm 2164.9$ | $2968.1 \pm 2174.6$ | $2460.6 \pm 1374.8$ | $2436.1 \pm 1337.6$ | $2397.3 \pm 1292.7$ |
> | NPO | $155.0 \pm 8.3$ | $135.7 \pm 12.3$ | $133.6 \pm 12.7$ | $128.7 \pm 13.5$ | $126.5 \pm 14.0$ | $116.6 \pm 20.3$ | $112.9 \pm 19.6$ | $109.2 \pm 18.9$ |
> | DPO | $49.0 \pm 1.6$ | $46.8 \pm 1.7$ | $46.9 \pm 1.7$ | $47.1 \pm 1.8$ | $47.0 \pm 1.6$ | $42.8 \pm 1.5$ | $41.5 \pm 1.3$ | $40.6 \pm 1.1$ |
> | IHL | $373.0 \pm 96.8$ | $341.4 \pm 84.1$ | $341.5 \pm 84.1$ | $341.3 \pm 83.6$ | $342.3 \pm 83.0$ | $353.6 \pm 90.4$ | $353.8 \pm 90.0$ | $352.1 \pm 89.5$ |
> | Retain | $10.5 \pm 0.2$ | $9.1 \pm 0.5$ | $9.0 \pm 0.5$ | $9.0 \pm 0.5$ | $8.9 \pm 0.7$ | $8.4 \pm 0.3$ | $8.6 \pm 0.3$ | $8.7 \pm 0.3$ |

---

### Author Response · Authors · 2025-11-20
**Common Response to All Reviewers**

We express our sincere gratitude to the reviewers for their constructive feedback and for highlighting the significance of the problem we address. We are deeply encouraged by reviewers' acknowledgment of various strengths in our work, namely:
- **Identifying Critical Gaps and Redirecting the Field [PTzk, 8y78, 7Aru, 1R1j].** FADE exposes a significant blind-spot in current practice by demonstrating that existing reference-specific metrics systematically overestimate unlearning efficacy, and redirects the field toward more rigorous evaluation standards.

- **Principled Foundation with Strong Empirical Validation [PTzk, 8y78, 7Aru, 1R1j].** FADE provides a principled approach of measuring unlearning efficacy via distributional equivalence with retain-only models (the gold standard for exact unlearning) and is supported by compelling empirical evidence.

- **Modality-agnostic Design and Applicability [PTzk, 7Aru].** FADE applies across multiple modalities (LLMs and T2I diffusion models) with a unified framework, demonstrating generality beyond existing metrics that are modality-specific.

In the following, we briefly discuss two concerns expressed by multiple reviewers, particularly on (1) **the practicality and compute-cost aspect of FADE**, and (2) **its robustness to varying decoding strategies**. More detailed responses to all comments can be found in the respective comments below.

- **FADE's practicality and compute cost [for PTzk and 8y78].** Regarding FADE's use of retain-only models: while they require additional compute, **retain-only models provide ground truth baselines essential for rigorous evaluation**. In addition, **this requirement is already standard practice**, as the widely-adopted TOFU benchmark's Forget Quality metric also requires retain-only models. We will open-source all checkpoints upon acceptance to eliminate redundant costs.
Regarding the computational cost of FADE, we provide runtime estimates of FADE in the responses below, but would also like to point out that (1) **evaluation accuracy well-justifies the cost** as inaccurate metrics risk misdirecting research far more costly than proper evaluation, (2) **FADE is highly parallelizable across multiple GPUs**, and (3) **our ablations suggest FADE remains robust with as few as 10 samples per prompt**, allowing practitioners to reduce costs while preserving key findings.

- **FADE's robustness to decoding strategy [for PTzk and 8y78].** We conducted comprehensive ablation studies to assess FADE's robustness across alternative decoding strategies, such as Nucleus Sampling and Beam Search, which may introduce estimation bias unlike multinomial sampling used in FADE as default. Results suggest (1) **FADE values remain consistent across different decoding strategies**, and (2) our core finding holds throughout: **regardless of decoding strategy, existing methods fail to approach the retain-only baseline**. Please refer to the reviewer-specific comments for detailed results.

---

### Meta-Review · Area_Chair_mW79 · 2026-01-12

**Summary:**

This paper highlights an existing problem in unlearning: unreliable metrics for verifying unlearning for generative models. The paper proposes Functional Alignment for Distributional Equivalence (FADE), which evaluates distributional alignment with retain-only oracles through bidirectional likelihood comparisons over generated samples.

The reviewers expressed following main concerns:
- While the FADE metric and evaluation are important to emphasize existing problems, FADE requires having a retain-only model. Thus, it is not clear whether it can be practically implemented (or a proxy of it) to improve the unlearning method itself.
- It is computationally expensive to compute the retain-only model. The Monte-Carlo style sampling and likelihood estimates can be expensive.
- The proposed metric can be sensitive to generation strategy, which raises concerns about consistency in evaluations.

Overall, I feel the assumption about having a retrained model is very strong and makes the proposed metric infeasible for any model where someone would want to perform unlearning rather than retraining.

**Reviewer Concerns:**

- On applications and computational complexity, authors argued that the proposed method is not meant to improve unlearning but merely to provide a good evaluation metric; and that the benefits outweigh the cost.

- Authors provided several additional experiments to who that FADE provides consistent evaluation for different generation strategies and with randomly initialized and retrained models.

**Reviewer Scores:**

The paper received 6,4,4,4 ratings.

I think the ratings would have remained the same or may have dropped further.

---

### Decision · Program_Chairs · 2026-01-26

Reject